# Does Trading Volume Drive Systemic Banks' Stock Return Volatility? Lessons from the Greek Banking System

Athanasios Tsagkanos [1,*], Konstantinos Gkillas [2] , Christoforos Konstantatos [1] and Christos Floros [3]

1   Department of Business Administration, University of Patras, University Campus—Rio, P.O. Box 1391, 26504 Patras, Greece; ckonstanta@upatras.gr
2   Department of Management Science & Technology, University of Patras, Megalou Aleksandrou 1, Koukouli, 26334 Patras, Greece; gillask@upatras.gr
3   Department of Accounting and Finance, Hellenic Mediterranean University, P.O. Box 1939, 72100 Crete, Greece; cfloros@hmu.gr
*   Correspondence: atsagkanos@upatras.gr

**Abstract:** The present research investigates the impact of trading volume on stock return volatility using data from the Greek banking system. For our analysis, the empirical study uses daily measures of volatility constructed from intraday data for the period 5 January 2001–30 December 2020. This period includes several market phases, such as the latest financial crisis, the European sovereign debt crisis and enforcement of restrictions on transactions owing to capital controls on the Athens Stock Exchange in June 2015. Based on the estimated quantile regressions, we find evidence of a direct impact of the trading volume on stock return volatility mainly in all quantiles. The findings extrapolated are of relevance and interest to financial (banking) analysts, policy makers and practitioners concerned with intraday data and volatility modeling.

**Keywords:** volatility; volume; realized measures; intraday data; extreme value theory; banks; Greece

## 1. Introduction

Financial market volatility is a factor of primary importance relevant to several issues in the field of finance, varying from asset management to risk management (Poon and Granger 2003). In this respect, market participants' main concerns are the nature and level of volatility. Considering the involvement of volatility in investment decision making, derivative pricing and financial market regulation, numerous approaches have been suggested in the relevant literature in terms of its estimation. A basic postulation is the change in volatility over time, and thereby such changes are subject to stochastic modeling. In the literature, it has been proven that stochastic volatility models outperform constant volatility models (see Hull and White 1987; Ghysels et al. 1996; Andersen and Lund 1997, among many others). In fact, there is evidence of non-stationarity in the variance (see, e.g., Cohen et al. 1972). Stochastic volatility models overcome the limitation observed in Black and Scholes' model, with the volatility being constant over time and immune to changes in the price level of the underlying asset. Parametric, semi-parametric and nonparametric parameters can estimate volatility, despite the fact that it is latent (see Asai et al. 2006; Maasoumi and McAleer 2008; Asai and McAleer 2011; Caporin and McAleer 2012, for an extensive elaboration on this topic), while statistical deductions of stochastic volatility models are mostly parametric. Cox et al. (1985) and Heston (1993) provided some typical examples of parametric estimation. Nevertheless, Alghalith (2012) stressed a number of considerable limitations such procedures display. A limited set of studies employed nonparametric approaches applying integrated volatility, some of which are those of Gkillas et al. (2021), Floros et al. (2020), Gkillas et al. (2020b), Vetter (2015),

Comte et al. (2010) and Renò (2006). Being a rudimentary concept in the literature in finance, stochastic volatility accounts for the inherent time-varying behavior of volatility and the co-dependence pattern attested among markets (see Mandelbrot 1963; Officer 1973; Shephard 2005, among others).

Further, it is important to see the relation of volatility with liquidity variables such as the trading volume. Low trading volume implies that the market is illiquid, resulting in high price volatility (Floros and Vougas 2007). The relationship between volatility and trading volume has been examined by financial economists for decades due to the fact that trading volume can be used for formulating a trading strategy (see Floros and Salvador 2016). This relation depends on the type of trader and the information (news arrival). Karpoff (1987) showed a positive relationship between trading volume and volatility, i.e., volume plays a significant role in market information. There are two leading theories on the correlation between trading volume and volatility: (i) The Mixture of Distributions Hypothesis (MDH) and the (ii) Sequential Information Arrival Hypothesis (SIAH). According to MDH (Clark 1973), prices and volume change when information arrives. One of the MDH implications is that, for a given time period, there is a positive correlation between volatility and volume. This is because both are positive functions of the rate of arrival of information during the time period (Harris 1987). On the other hand, the SIAH argues that each trader observes the information sequentially (Floros and Vougas 2007). It supports a dynamic relationship which gives useful information about the trading volume and forecasts of volatility (Copeland 1976). Both MDH and SIAH imply a positive and contemporaneous relationship between daily trading volume and volatility (see Floros and Salvador 2016).

In recent years, the banking system has been adversely affected by globalization and general instability (economic, political and social). It has been noticed that banking institutions are constantly reorganizing and modifying due to financial pressures. As a result, banking institutions are operating in an environment of uncertainty due to the global financial crisis. Moreover, the global financial crisis presented in a different form in the economic environment at the international level (Gkillas et al. 2019a, 2020a, 2020d; Vortelinos et al. 2017). The crisis broke out in August 2007 in the US, and in 2008, the most well-known financial services company collapsed. Due to the collapse of the Lehman Brothers, the financial crisis began to spread worldwide, both in Europe and consequently in Greece. The crisis has had an impact on all countries, regardless of size, power, economic situation and the policy pursued by each country, affecting the efficiency of banking institutions (Musa et al. 2020; Balcerzak et al. 2017). All countries had to adapt to the new economic conditions (Ozturk and Sozdemir 2015). The crisis in the Greek economy appeared at the end of 2008, recording a deficit of 9.8% of the GDP and public debt of 112.9% of the GDP (Bank of Greece 2014). No one had understood how critical and serious the situation was in the Greek economy and especially in the banking sector because at the beginning of the crisis, Greece was little affected by the American financial crisis (Pagoulatos and Triantopoulos 2009). The following year began to become worse for Greece because economic problems such a public debt, deficits and low competitiveness, which the country was trying to cover through economic growth, came to the surface (Mitsopoulos and Pelagidis 2011). Due to these problems, the banking sector was "hit" first by the global crisis.

The Greek financial system exhibits the following characteristics: (i) it is an oligopolistic structure of the domestic financial system; (ii) it operates in a small emerging market with an oligopolistic bank-dominated financial system; (iii) it has poorly educated investors in terms of finance in combination with invisibility of expenses; (iv) the Athens Stock Exchange (ASE) is small-sized, displaying illiquidity. All the above-mentioned features render the Greek banking industry a challenging case that is well worth studying. Banking institutions constitute the basic market players due to the fact that their capitalization stands for a significant proportion of the total Athens Stock Exchange (ASE) capitalization and because of their possessing a considerable portion of further listed companies either

directly or indirectly, namely, by means of their mutual fund companies (Babalos et al. 2009). However, it was the political instability and the high sovereign debt which affected the real economy, imposing restrictions on cash withdrawals and capital transfers on 28 June 2015 (Gkillas Gillas).

Seldom are capital restraints enforced with regard to financial crises and far-reaching financial instability, with Iceland and Cyprus being the latest cases of capital restrictions imposed in 2008 and 2013, respectively (Gkillas Gillas). Contradictory opinions are found in the existing literature concerning outflow restrictions as a crisis management tool (Magud et al. 2011). Two viewpoints have been voiced on this issue. According to the first viewpoint, the enforcement of capital controls as a short-term tool for the stabilization of the economy is necessary when common measures have not been effective, and panic and extreme financial instability plague the economy (Krugman 1998). Embracing this viewpoint, Bhagwati (1998) and Rodrik (2000) stressed the need for outflow restrictions, particularly over the short-term horizon, whereas in the study of Eichengreen and Rose (2014) on capital controls in modern economies, financial capital restrictions were shown to be persistent and not short term due to their long duration and their resemblance to phenomena including international trade policy regulations and exchange rate regimes. In the same vein, it has been established that cronyism can be retained by the imposed restrictions on capital flows (Johnson and Mitton 2003).

According to the other viewpoint, a rise in inflation can be noticed by the restriction of capital outflows in case investors are unable to maintain their funds in foreign assets (Alesina et al. 1993). The inefficiency, administrative cost and susceptibility to corruption that capital restrictions display have been shown in the studies of Edwards (1999) and Schmidt (2013). The tendency of capital inflows or outflows to be volatile circumstantially and the difficulty inherent in the decision to put into force policies on restrictions have also been stressed (Broto et al. 2011). Fernández et al. (2013) questioned the wisdom of applying capital controls in periods of real economic crisis. Furthermore, they cannot mitigate output and exchange rate fluctuations. On top of that, the deceleration of the US economic recovery and the detraction from the efficiency of monetary policy tools were the function of the capital controls imposed in the 1930s, as Mitchener and Wandschneider (2015) claimed. Rather, in case of a lack of strict and austere regulation of capital controls, more freedom to affect risk metrics is provided to managers (Bryce et al. 2015); still, this may be the impression of financial stability and safety given to regulators, which may indeed be misguiding.

As volatility is central to many issues in finance, it is reported in asset and risk management for policy issues. In this paper, we study the effect of trading volume on stock return volatility of the Greek banking system in a sensitive period that includes, among others, the European sovereign debt crisis, the Greek crisis and the restrictions on transactions due to capital controls in 2015. Greek banking uncertainty was related to the capital controls imposed in Greece in June 2015 in an attempt to avoid an uncontrolled bank run; see Samitas and Polyzos (2016). The research's aim is to investigate the impact of trading volume on banks' stock return volatility considering a large dataset from four systemic banking institutions in Greece, a country which suffered a lot from the 2008 financial crisis according to official statistics and studies. In particular, Stournaras (2019) reported a high public debt ratio, a high NPL ratio and high long-term unemployment. The Greek economy returned to growth after several years of financial assistance programs, and therefore it is still an important case study (for more information about the Greek banking system and its relationship with the economy, see Floros 2020).

The remainder of this paper is organized as follows. Section 2, first, discusses the data used in this study and the descriptive statistics. Section 3 presents the methodological approach to construct the volatility estimators and estimate the impact of trading volume on stock return volatility. Section 4 provides the empirical results, Section 5 discuss the results Finally, Section 6 concludes the paper.

## 2. Data and Descriptive Statistics

This section presents the data used in this study. This study employs the bank equity prices, both open and closing price series, including the volume for the systemic banking institutions in Greece, considering (i) Alpha Bank (ALPHA), (ii) Eurobank (EUROB), (iii) National Bank of Greece (ETE) and (iv) Piraeus Bank (TPEIR). Those banking institutions are considered as the larger banks in Greece belonging to, among the 24 larger banks, the European Monetary Union (see, Gkillas et al. 2019b). The dataset consists of daily values of the aforementioned variables and covering a sample period from 5 January 2001 to 30 December 2020 (4934 observations). The data were retrieved from the Thomson Reuters database. Our sample period includes different market phases, such as major booms and crashes (e.g., the European sovereign debt crisis, Greek crisis and restrictions on transactions due to capital controls on the Athens Stock Exchange).

Table 1 and Panel A report basic statistics of the closing prices of four Greek banking institutions, Alpha Bank, National Bank of Greece, Eurobank and Piraeus Bank, which are denoted by ALPHA, EUROB, ETE and TPEIR, respectively. In this table, the mean of the closing prices of ALPHA is equal to 244.5589, with a standard deviation equal to 261.3332, a maximum of 921.8120 and a minimum of 0.4160. As for the closing prices of EUROB, the mean is equal to 8498.103, with a standard deviation equal to 9726.920, a maximum of 33,829.6 and a minimum of 0.2830. As for the ETE, the mean of closing prices is equal to 14,462.05, with a standard deviation equal to 17,126.83, a maximum of 62,776.8 and a minimum of 0.8500. As for the TPEIR, the mean of closing prices is equal to 3363.702, with a standard deviation equal to 4061.842, a maximum of 14,773.9 and a minimum of 0.0810. In all cases, the positive skewness indicates that the tail of the left side of the distribution is longer. Moreover, the kurtosis on ALPHA and EUROB is below 3; however, on ETE and TPEIR, it is close to 3, indicating that the distributions of closing prices are close to normal distribution. Lastly, the normality of all series under consideration was tested with the Jarque–Bera test. In all cases, the null hypothesis of normality is rejected at a 1% significance level.

**Table 1.** Descriptive statistics.

|  | **ALPHA** | **EUROB** | **ETE** | **TPEIR** |
|---|---|---|---|---|
|  | Panel A. Closing prices | | | |
| Mean | 244.5589 | 8498.103 | 14,462.05 | 3363.702 |
| Median | 124.4860 | 4200.00 | 8476.54 | 1769.41 |
| Maximum | 921.8120 | 33,829.6 | 62,776.8 | 14,773.9 |
| Minimum | 0.4160 | 0.2830 | 0.8500 | 0.0810 |
| Std. Dev. | 261.3332 | 9726.920 | 17,126.830 | 4061.842 |
| Skewness | 0.5943 | 0.776571 | 1.074902 | 1.106899 |
| Kurtosis | 1.7999 | 2.2307 | 3.0252 | 3.1090 |
| Jarque–Bera | 586.8475 *** | 617.9354 *** | 950.2674 *** | 1010.399 *** |
| Jarque–Bera prob. | [0.0000] | [0.0000] | [0.0000] | [0.0000] |
|  | Panel B. Trading volume | | | |
| Mean | 5,079,130 | 2,884,767 | 2,657,638 | 885,629.5 |
| Median | 3,520,000 | 578,290 | 140,410 | 245,770 |
| Maximum | $1.43 \times 10^8$ | $3.73 \times 10^8$ | $2.37 \times 10^8$ | $2.84 \times 10^8$ |
| Minimum | 46,880 | 12,640 | 1940 | 1230 |
| Std. Dev. | 6,215,654 | 13,317,116 | 8,225,758 | 4,442,894 |
| Skewness | 6.354414 | 14.14456 | 13.97726 | 53.08828 |
| Kurtosis | 88.58104 | 283.4652 | 314.7657 | 3352.597 |
| Jarque–Bera | 1,539,856 *** | 16,345,788 *** | 20,130,660 *** | $2.31 \times 10^9$ *** |
| Jarque–Bera prob. | [0.0000] | [0.0000] | [0.0000] | [0.0000] |
| Obs | 4934 | 4934 | 4934 | 4934 |

Notes: This table reports descriptive statistics for the closing prices and trading volume of four banking institutions in Greece, Alpha Bank (ALPHA), Eurobank (EUROB), National Bank of Greece (ETE) and Piraeus Bank (TPEIR). Panel A refers to closing prices, while Panel B refers to trading volume. The following statistics are given: mean, median, maximum, minimum, standard deviation (Std. Dev), skewness, kurtosis, Jarque–Bera test and the number of observations. The null hypothesis that the series is normally distributed is also tested by the Jarque–Bera test. *** indicates a rejection of the null hypothesis of normality at a 1% significance level.

Table 1 and Panel B report basic statistics of the trading volume of the Greek banking institutions. As for ALPHA, the mean of the trading volume is equal to 5,079,130 with standard deviation equal to 6,215,654. The mean of the trading volume of ETE is equal to 2,884,767 and the standard deviation is equal to 13,317,116. Looking at EUROB, we can observe that the mean of the trading volume is equal to 2,657,638, while the standard deviation is equal to 8,225,758. It is shown that the mean and the standard deviation of the trading volume in TPEIR are equal to 885,629.5 and 4,442,894, respectively. The skewness on ALPHA, ETE, EUROB and TPEIR is observed to be positive skewness, indicating a longer left tail of the distribution. Moreover, the kurtosis in all cases is greater than 3, indicating that the distributions of trading volumes have fat tails. Lastly, we tested the normality of all series under consideration with the Jarque–Bera test. In all cases, the null hypothesis of normality is rejected at a 1% significance level.

### 3. Methods

We made the following assumptions that the price $P$ follows a geometric Brownian motion such that log-price $p = \ln(P)$ follows a Brownian motion with zero drift and diffusion $\sigma$, as follows:

$$dp_t = \sigma dB_t \tag{1}$$

where $B_t$ stands for Brownian motion.

In addition, we assumed that the diffusion parameter $\sigma$ is unchanged during the day, but it should be noted that it changes from day to day. One day is defined as a unit of time. We observed that the diffusion parameter in Equation (1) identifies with the daily standard deviation of returns declaring normalization and so these quantities do not require distinction from us. We set the opening price of the day $O$, the closing price of the day $C$, the highest price of the day $H$ and the lowest price of the day $L$ to calculate open-to-close, open-to-high and open-to-low returns with the following equations:

$$c = \ln(C) - \ln(O) \tag{2}$$

$$h = \ln(H) - \ln(O) \tag{3}$$

$$l = \ln(L) - \ln(O) \tag{4}$$

where, $c$ stands for open-to-close returns, $h$ stands for open-to-high returns and $l$ stands for open-to-low returns.

Daily return $c$ is a random variable which follows the normal distribution with the mean 0 and variance (volatility) $\sigma^2$:

$$c \sim N\left(0, \sigma^2\right) \tag{5}$$

Now, we aimed to estimate (unobservable) volatility $\sigma^2$ from variables $c$, $h$ and $l$ that have been observed. It is known that $c^2$ is an unbiased estimator of $\sigma^2$,

$$E\left(c^2\right) = \sigma^2 \tag{6}$$

and the first volatility estimator (subscript $s$ stands for "simple") is given by

$$\hat{\sigma_s^2} = c^2 \tag{7}$$

Given that the simple estimator is too noisy, we would prefer to have a better one. It appears that the varieties between the upper and lowest price points exhibit much information, mainly concerning volatility rather than the closing price. Additional volatility information is obtained from high and low prices. The distribution of the range $d \equiv h - l$ (defined as the difference between the highest and lowest prices) of Brownian motion is

well known (Feller 1951). We defined as $P(x)$ the probability that $d \leq x$ is valid during the day,

$$P(x) = \sum_{n=1}^{\infty} (-1)^{n+1} n \left\{ Erfc\left(\frac{(n+1)x}{\sqrt{2}\sigma}\right) - 2Erfc\left(\frac{nx}{\sqrt{2}\sigma}\right) + Erfc\left(\frac{(n-1)x}{\sqrt{2}\sigma}\right) \right\} \quad (8)$$

where

$$Erfc = 1 - Erf(x) \quad (9)$$

where $Erf(x)$ stands for the error function. We used the distribution of Parkinson (1980) to calculate (for $p \geq 1$)

$$E\left(d^p\right) = \frac{4}{\sqrt{\pi}} \Gamma\left(\frac{p+1}{2}\right) \left(1 - \frac{4}{2^p}\right) \zeta(p-1)\left(2\sigma^2\right) \quad (10)$$

where $\Gamma(x)$ stands for the gamma function and $\zeta(x)$ stands for the Riemann zeta function. Specifically, for $p = 1$,

$$E(d) = \sqrt{8\pi}\sigma \quad (11)$$

and for $p = 2$,

$$E\left(d^2\right) = 4\ln(2)\sigma^2 \quad (12)$$

According to Equation (12), a new volatility estimator is proposed, by Garman and Klass (1980):

$$\hat{\sigma}_p^2 = \frac{(h-l)^2}{4ln2} \quad (13)$$

Garman and Klass (1980) came to realize that the estimator base lies on quantity $h - l$ alone and thus an estimator utilizing the totality of information available would necessarily be more precise. Due to the fact that searching for the minimum variance estimator which is based on $c$, $h$ and loved seems to be an infinite dimensional problem, this problem is restricted to analytical estimators, that is, the ones that may be described as an analytical function of $c$, $h$ and loved. They seem to conclude that the minimum variance analytical estimator is provided by the following equation:

$$\hat{\sigma^2_{GKprecise}} = 0.511(h-l)^2 - 0.019(c(h+l) - 2hl) - 0.383c^2 \quad (14)$$

The right term (cross-products) is negligible, and consequently, it is suggested that we overlook it and use a more practical estimator:

$$\hat{\sigma}_{GK}^2 = 0,5(h-l)^2 - (2ln2-1)c^2 \quad (15)$$

Following Garman and Klass (1980), the Garman–Klass volatility estimator (GK), referring to Equation (15), is a further advantage over the estimator in Equation (14), which is explained as the optimal (smallest variance) combination of simple and Parkinson volatility estimators (see also Floros 2009).

According to Meilijson (2009), another estimator is obtained, outside the class of analytical estimators, which has the smallest variance. The construction of the estimator is mentioned below:

$$\hat{\sigma}_M^2 = 0.274\sigma_1^2 + 0,16\sigma_s^2 + 0.365\sigma_3^2 + 0.2\sigma_4^2 \quad (16)$$

where

$$\sigma_1^2 = 2\left[(h'-c')^2 + l'\right] \quad (17)$$

$$\sigma_3^2 = 2(h'-c'-l')c' \quad (18)$$

$$\sigma_4^2 = -\frac{(h'-c')l'}{2ln2 - \frac{5}{4}} \quad (19)$$

where $c' = c$, $h' = h$, $l' = l$ *if* $c > 0$ *and* $c' = -c$, $h' = -l$, $l' = -h$ *if* $c < 0$.

This estimator is not analytical because it uses a different formula when $c > 0$ than (for days) when $c < 0$.

According to Rogers and Satchell (1991), the following estimator

$$\hat{\sigma}^2_{RS} = h(h - c) + l(l - c) \tag{20}$$

allows for arbitrary drift.

*Impact of Volatility*

Using quantile nonparametric simple regression, the present study investigates the impact of trading volume on volatility of four Greek banking institutions, Alpha Bank, National Bank of Greece, Eurobank and Piraeus Bank, for daily frequency (Gkillas et al. 2020c). For daily frequency, we use the following quantiles: 0.05, 0.20, 0.40, 0.60, 0.80 and 0.95.

According to Koenker and Mizera (2004), the present study estimates the nonparametric series (volatility estimators) quantile regression. In particular, the coefficient vector $\delta$, as the direct impact of trading volume in each Greek bank on volatility, is estimated. In Equation (21), the definition of quantile regression is given:

$$\sigma^{i,q}_{t,j} = a + \delta^i_j V_{t-1,j} + u^{i,q}_{t,j} \ , \ q \in (0,1) \tag{21}$$

where $\sigma^{i,q}_{t,j}$ stands for the volatility at quantile $q$ of $i$ volatility estimator, where $i = 1, \ldots, 4$ is any of the following volatility series: $\hat{\sigma}^2_p$, $\hat{\sigma}^2_{GK}$, $\hat{\sigma}^2_M$ and $\hat{\sigma}^2_{RS}$ of $j$ banking institution at time $t$, where $j = 1, \ldots, 4$, representing the following banking institutions: (i) Alpha Bank, (ii) Eurobank, (iii) National Bank of Greece and (iv) Piraeus Bank. Additionally, $V_{t,j}$ stands for the explanatory variable of trading volume of $j$ banking institution at time $t$.

## 4. Empirical Results

Tables 1–4 report the impact of trading volume on stock return volatility considering four volatility estimators such as (i) $\sigma^2_p$, (ii) $\sigma^2_{GK}$, (iii) $\sigma^2_M$ and (iv) $\sigma^2_{RS}$, for Alpha Bank (Table 1), National Bank of Greece (Table 2), Eurobank (Table 3) and Piraeus Bank (Table 4).

Table 2 reports the direct impact of the trading volume of ALPHA ($\delta_{ALPHA}$) on stock return volatility. At the 0.05 quantile, it is observed that the direct impact ($\delta_{ALPHA}$) of trading volume on volatility estimator $\sigma^2_p$ is statistically significant at 1%, which equals $5.159 \times 10^{-13}$. At the 0.20 and 0.40 quantiles, it is observed that the direct impact ($\delta_{ALPHA}$) of trading volume on volatility estimator $\sigma^2_p$ is also statistically significant at 1% and equals $4.079 \times 10^{-12}$ and $1.491 \times 10^{-11}$, respectively. Moreover, the impact of trading volume on volatility estimator $\sigma^2_p$ equals $2.868 \times 10^{-11}$, $5.706 \times 10^{-11}$ and $1.439 \times 10^{-10}$ at the 0.60, 0.80 and 0.95 quantiles, respectively, and is statistically significant at 1% in all cases.

Subsequently, the tables will be annotated based on the highest and lowest absolute values for each quantile. In Table 2, at the 0.05 quantile, it is observed that the highest value of trading volume impact on $\sigma^2_M$ is statistically significant at 1% and equals 1.906e-10, and the lowest value on $\sigma^2_{RS}$ equals $3.817 \times 10^{-13}$, with statistical significance at 1%. At the 0.20 quantile, it is observed that the highest value of trading volume impact on $\sigma^2_M$ equals $1.065 \times 10^{-10}$, and the lowest value on $\sigma^2_{RS}$ equals $1.930 \times 10^{-12}$, with statistical significance at 1%. Moreover, at the 0.40 quantile, the highest value of trading volume impact mentioned on $\sigma^2_p$ and the lowest value on $\sigma^2_M$ equal $1.491 \times 10^{-11}$ and $6.932 \times 10^{-12}$, respectively, with statistical significance at 1%. At the 0.60 quantile, it is observed that the highest value of trading volume impact on $\sigma^2_M$ equals $1.422 \times 10^{-10}$, with statistical significance at 1%, while the lowest value of trading volume impact on $\sigma^2_p$ equals $2.868 \times 10^{-11}$, with statistical significance at 1%. In addition, at the 0.80 (0.95) quantile, the highest value of trading volume impact observed on $\sigma^2_M$ is equal to $1.580 \times 10^{-10}$ ($2.829 \times 10^{-10}$), while the

lowest value on $\sigma^2_{GK}$ is equal to $5.398 \times 10^{-11}$ ($1.275 \times 10^{-10}$), with statistical significance at 1%.

**Table 2.** Trading volume impact on Alpha Bank's stock return volatility.

| $q$ | $\sigma^2_p$ | $\sigma^2_{GK}$ | $\sigma^2_M$ | $\sigma^2_{RS}$ |
|---|---|---|---|---|
| **0.05** | $5.159 \times 10^{-13}$ *** | $-1.906 \times 10^{-10}$ *** | $9.546 \times 10^{-13}$ *** | $-3.817 \times 10^{-13}$ *** |
|  | $(1.027 \times 10^{-13})$ | $(6.986 \times 10^{-12})$ | $(1.001 \times 10^{-13})$ | $(6.414 \times 10^{-14})$ |
| **0.20** | $4.079 \times 10^{-12}$ *** | $-1.065 \times 10^{-10}$ *** | $3.501 \times 10^{-12}$ *** | $1.930 \times 10^{-12}$ *** |
|  | $(1.782 \times 10^{-13})$ | $(1.949 \times 10^{-12})$ | $(1.567 \times 10^{-13})$ | $(1.462 \times 10^{-13})$ |
| **0.40** | $1.491 \times 10^{-11}$ *** | $-6.932 \times 10^{-12}$ *** | $1.148 \times 10^{-11}$ *** | $9.424 \times 10^{-12}$ *** |
|  | $(2.095 \times 10^{-13})$ | $(7.263 \times 10^{-13})$ | $(3.726 \times 10^{-13})$ | $(3.648 \times 10^{-13})$ |
| **0.60** | $2.868 \times 10^{-11}$ *** | $-1.422 \times 10^{-10}$ *** | $2.798 \times 10^{-11}$ *** | $2.928 \times 10^{-11}$ *** |
|  | $(8.486 \times 10^{-13})$ | $(3.091 \times 10^{-13})$ | $(8.698 \times 10^{-13})$ | $(5.456 \times 10^{-13})$ |
| **0.80** | $5.706 \times 10^{-11}$ *** | $1.580 \times 10^{-10}$ *** | $5.398 \times 10^{-11}$ *** | $5.578 \times 10^{-11}$ *** |
|  | $(1.020 \times 10^{-12})$ | $(1.168 \times 10^{-11})$ | $(1.353 \times 10^{-12})$ | $(1.718 \times 10^{-12})$ |
| **0.95** | $1.439 \times 10^{-10}$ *** | $2.829 \times 10^{-10}$ *** | $1.275 \times 10^{-10}$ *** | $1.423 \times 10^{-10}$ *** |
|  | $(6.986 \times 10^{-14})$ | $(2.027 \times 10^{-11})$ | $(8.509 \times 10^{-13})$ | $(4.765 \times 10^{-12})$ |

Notes: This table reports the direct impact of trading volume on Alpha Bank's stock return volatility, considering the volatility estimators (i) $\sigma^2_p$, (ii) $\sigma^2_{GK}$, (iii) $\sigma^2_M$ and (iv) $\sigma^2_{RS}$. *** indicates statistical significance at 1%.

As for Tables 3–5, the discussion of higher and lower impact values is based on each quantile. Table 3 reports the direct impact of the trading volume of EUROB ($\delta_{EUROB}$) on stock return volatility. In Table 3, it is observed that the highest value of trading volume impact has a range from $1.038 \times 10^{-11}$ to $4.985 \times 10^{-10}$, found at 0.40 on $\sigma^2_p$ and at the 0.95 quantile on $\sigma^2_{GK}$, respectively, while the lowest value impact of trading volume with a range from $1.554 \times 10^{-13}$ to $2.937 \times 10^{-11}$ is found at the 0.20 quantile on $\sigma^2_{RS}$ and at the 0.80 quantile on $\sigma^2_M$, respectively. This indicates that the higher impact is observed mainly on $\sigma^2_p$ at quantiles of 0.40 and 0.60 and on $\sigma^2_{GK}$ at lower and upper quantiles (0.05, 0.20, 0.80 and 0.95), while the lowest impact is observed mainly on $\sigma^2_{RS}$ in most quantiles. Moreover, mainly all direct impacts of the trading volume of EUROB ($\delta_{EUROB}$) on EUROB stock return volatility are statistically significant at 1%, with the exception of the volume impact on the $\sigma^2_p$ volatility estimator at the 0.05 quantile.

**Table 3.** Trading volume impact on Eurobank's stock return volatility.

| $q$ | $\sigma^2_p$ | $\sigma^2_{GK}$ | $\sigma^2_M$ | $\sigma^2_{RS}$ |
|---|---|---|---|---|
| **0.05** | $4.863 \times 10^{-14}$ | $-4.679 \times 10^{-11}$ *** | $5.636 \times 10^{-13}$ *** | $-1.767 \times 10^{-13}$ *** |
|  | $(2.469 \times 10^{-13})$ | $(1.272 \times 10^{-12})$ | $(2.136 \times 10^{-13})$ | $(3.147 \times 10^{-14})$ |
| **0.20** | $3.186 \times 10^{-12}$ *** | $-7.986 \times 10^{-11}$ *** | $3.963 \times 10^{-12}$ *** | $1.554 \times 10^{-13}$ ** |
|  | $(6.950 \times 10^{-14})$ | $(6.708 \times 10^{-13})$ | $(4.398 \times 10^{-14})$ | $(7.657 \times 10^{-14})$ |
| **0.40** | $1.038 \times 10^{-11}$ *** | $-5.357 \times 10^{-12}$ *** | $6.365 \times 10^{-12}$ *** | $3.349 \times 10^{-12}$ *** |
|  | $(1.824 \times 10^{-13})$ | $(6.561 \times 10^{-13})$ | $(8.656 \times 10^{-14})$ | $(8.079 \times 10^{-14})$ |
| **0.60** | $2.361 \times 10^{-11}$ *** | $1.172 \times 10^{-12}$ *** | $1.693 \times 10^{-11}$ *** | $1.200 \times 10^{-11}$ *** |
|  | $(1.557 \times 10^{-12})$ | $(1.272 \times 10^{-12})$ | $(4.240 \times 10^{-13})$ | $(5.575 \times 10^{-13})$ |
| **0.80** | $5.655 \times 10^{-11}$ *** | $1.328 \times 10^{-10}$ *** | $2.937 \times 10^{-11}$ *** | $3.091 \times 10^{-11}$ *** |
|  | $(1.235 \times 10^{-12})$ | $(8.255 \times 10^{-13})$ | $(2.37 \times 10^{-13})$ | $(1.213 \times 10^{-12})$ |
| **0.95** | $6.052 \times 10^{-11}$ *** | $4.985 \times 10^{-10}$ *** | $3.507 \times 10^{-11}$ *** | $-1.575 \times 10^{-11}$ *** |
|  | $(8.959 \times 10^{-13})$ | $(2.648 \times 10^{-11})$ | $(9.006 \times 10^{-12})$ | $(2.230 \times 10^{-12})$ |

Notes: This table reports the direct impact of the trading volume on Eurobank's stock return volatility, considering the volatility estimators (i) $\sigma^2_p$, (ii) $\sigma^2_{GK}$, (iii) $\sigma^2_M$ and (iv) $\sigma^2_{RS}$. *** and ** indicate statistical significance at 1%, and 5%, respectively.

Table 4 reports the direct impact of the trading volume of ETE ($\delta_{ETE}$) on stock return volatility. In Table 4, it is observed that the highest value of trading volume impact is observed mainly on the $\sigma^2_{GK}$ estimator with a range from $9.703 \times 10^{-13}$ to $9.816 \times 10^{-11}$, while the lowest value impact of $1.594 \times 10^{-13}$ to $9.745 \times 10^{-12}$ is found on $\sigma^2_p$ in lower

quantiles (0.05, 0.20 and 0.40) and on $\sigma_M^2$ in upper quantiles (0.60, 0.80 and 0.95). Furthermore, all direct impacts of the trading volume of ETE ($\delta_{ETE}$) on ETE stock return volatility presented in Table 4 are statistically significant at 1%.

**Table 4.** Trading volume impact on National Bank of Greece's stock return volatility.

| q | $\sigma_p^2$ | $\sigma_{GK}^2$ | $\sigma_M^2$ | $\sigma_{RS}^2$ |
|---|---|---|---|---|
| **0.05** | $1.084 \times 10^{-12}$ *** | $-7.992 \times 10^{-11}$ *** | $1.477 \times 10^{-12}$ *** | $1.192 \times 10^{-12}$ *** |
| | $(6.871 \times 10^{-15})$ | $(1.684 \times 10^{-12})$ | $(7.233 \times 10^{-15})$ | $(0.0000)$ |
| **0.20** | $8.156 \times 10^{-13}$ *** | $-1.928 \times 10^{-11}$ *** | $1.212 \times 10^{-12}$ *** | $1.211 \times 10^{-12}$ *** |
| | $(6.441 \times 10^{-15})$ | $(5.083 \times 10^{-14})$ | $(6.846 \times 10^{-15})$ | $(0.0000)$ |
| **0.40** | $2.973 \times 10^{-13}$ *** | $-8.835 \times 10^{-13}$ *** | $7.101 \times 10^{-13}$ *** | $9.703 \times 10^{-13}$ *** |
| | $(0.0000)$ | $(3.913 \times 10^{-14})$ | $(0.0000)$ | $(0.0000)$ |
| **0.60** | $8.870 \times 10^{-13}$ *** | $-3.416 \times 10^{-11}$ *** | $-1.594 \times 10^{-13}$ *** | $1.003 \times 10^{-12}$ *** |
| | $(4.382 \times 10^{-14})$ | $(5.322 \times 10^{-13})$ | $(2.733 \times 10^{-14})$ | $(0.0000)$ |
| **0.80** | $-1.563 \times 10^{-12}$ *** | $-2.686 \times 10^{-11}$ *** | $-1.375 \times 10^{-12}$ *** | $-1.521 \times 10^{-12}$ *** |
| | $(1.036 \times 10^{-13})$ | $(6.336 \times 10^{-13})$ | $(1.371 \times 10^{-13})$ | $(1.300 \times 10^{-13})$ |
| **0.95** | $1.073 \times 10^{-11}$ *** | $-9.816 \times 10^{-11}$ *** | $-9.745 \times 10^{-12}$ *** | $-1.018 \times 10^{-11}$ *** |
| | $(1.331 \times 10^{-12})$ | $(4.887 \times 10^{-12})$ | $(7.772 \times 10^{-13})$ | $(5.441 \times 10^{-13})$ |

Notes: This table reports the direct impact of the trading volume on National Bank of Greece's stock return volatility, considering the volatility estimators (i) $\sigma_p^2$, (ii) $\sigma_{GK}^2$, (iii) $\sigma_M^2$ and (iv) $\sigma_{RS}^2$. ***, indicate statistical significance at 1%.

Table 5 reports the direct impact of the trading volume of TPEIR ($\delta_{TPEIR}$) on stock return volatility. The results in Table 5 are mainly similar to Table 3 and the direct impact of EUROB more specifically, with the highest value of trading volume impact with a range from $2.667 \times 10^{-11}$ to $3.585 \times 10^{-9}$ observed mainly on $\sigma_p^2$ and $\sigma_{GK}^2$. However, the lowest value is found mainly on $\sigma_{RS}^2$ with a range from $7.937 \times 10^{-12}$ to $9.619 \times 10^{-10}$ at the 0.20 and 0.95 quantiles, respectively. Additionally, all direct impacts of the trading volume of TPEIR ($\delta_{TPEIR}$) on TPEIR stock return volatility presented in Table 5 are statistically significant at 1%. In conclusion, according to Tables 3–5, we mainly observe that the trading volume of all Greek banks has a direct effect on stock return volatility with statistical significance at 1%.

**Table 5.** Trading volume impact on Piraeus Bank's stock return volatility.

| q | $\sigma_p^2$ | $\sigma_{GK}^2$ | $\sigma_M^2$ | $\sigma_{RS}^2$ |
|---|---|---|---|---|
| **0.05** | $1.226 \times 10^{-11}$ *** | $-1.244 \times 10^{-9}$ *** | $1.069 \times 10^{-11}$ *** | $8.389 \times 10^{-12}$ *** |
| | $(8.770 \times 10^{-14})$ | $(2.404 \times 10^{-11})$ | $(0.0000)$ | $(0.0000)$ |
| **0.20** | $1.345 \times 10^{-11}$ *** | $-2.053 \times 10^{-10}$ *** | $1.031 \times 10^{-11}$ *** | $7.937 \times 10^{-12}$ *** |
| | $(8.109 \times 10^{-15})$ | $(4.454 \times 10^{-12})$ | $(7.100 \times 10^{-15})$ | $(0.0000)$ |
| **0.40** | $2.667 \times 10^{-11}$ *** | $-7.239 \times 10^{-19}$ | $1.792 \times 10^{-11}$ *** | $9.977 \times 10^{-12}$ *** |
| | $(1.710 \times 10^{-12})$ | $(8.105 \times 10^{-12})$ | $(1.166 \times 10^{-12})$ | $(7.945 \times 10^{-13})$ |
| **0.60** | $9.655 \times 10^{-11}$ *** | $1.320 \times 10^{-10}$ *** | $9.037 \times 10^{-11}$ *** | $7.452 \times 10^{-11}$ *** |
| | $(3.973 \times 10^{-12})$ | $(4.914 \times 10^{-13})$ | $(3.510 \times 10^{-12})$ | $(2.754 \times 10^{-12})$ |
| **0.80** | $2.973 \times 10^{-10}$ *** | $9.073 \times 10^{-11}$ *** | $2.765 \times 10^{-10}$ *** | $2.660 \times 10^{-10}$ *** |
| | $(9.711 \times 10^{-12})$ | $(7.569 \times 10^{-13})$ | $(9.465 \times 10^{-12})$ | $(7.025 \times 10^{-12})$ |
| **0.95** | $1.054 \times 10^{-9}$ *** | $3.585 \times 10^{-9}$ *** | $1.033 \times 10^{-9}$ *** | $9.619 \times 10^{-10}$ *** |
| | $(3.947 \times 10^{-11})$ | $(9.948 \times 10^{-11})$ | $(2.634 \times 10^{-11})$ | $(3.501 \times 10^{-11})$ |

Notes: This table reports the direct impact of the trading volume on Piraeus Bank's stock return volatility, considering the volatility estimators (i) $\sigma_p^2$, (ii) $\sigma_{GK}^2$, (iii) $\sigma_M^2$ and (iv) $\sigma_{RS}^2$. *** and * indicate statistical significance at 1% and 10%, respectively.

## 5. Discussion

Our results are in line with Darrat et al. (2003), who found significant lead–lag relations between the trading volume and return volatility in all stocks in the Dow Jones industrial average (DJIA) index. However, Chuang et al. (2012) and Andersen (1996) provided

evidence of a contemporaneous relation between stock trading volume and returns. Chuang et al. (2012), by employing bivariate GARCH models, simultaneously investigated the relation and causal relation between trading volume and stock returns, finding evidence that the contemporaneous relation between stock returns and trading volume and the causal relation between stock returns and trading volume are significant among the Asian markets. More specifically, Andersen (1996) mentioned that "The contemporaneous relation is derived from a stylized microstructure framework in which informational asymmetries and liquidity needs motivate trade in response to the arrival of new information." Taking into consideration the results of the present paper which reveal the significant relation between trading volume and volatility, and the results of Gkillas and Longin (2018), which concluded that a significant correlation between returns and trading volume exists, this relation acts as an amplification mechanism to extremely volatile periods in the ASE (Athens Stock Exchange) when capital controls are imposed. Koulakiotis et al. (2015) highlighted that the trading volume acted as a moderator for the conditional price volatility in the ASE over the harsh times of the Greek crisis as well as prior to the enforcement of capital controls, indicating that increasing the liquidity via trading volume may improve the climate and deter investors from panic and extremely volatile periods. A possible automatic short-term or intraday circuit breaker mechanism when there is a downward movement of the market in excess of a particular quantile (see Booth and Broussard 1998; Gkillas and Longin 2018) could benefit the real economy to avoid long-term ineffective distortions by imposing capital controls in June 2015.

## 6. Conclusions

Following the filing for bankruptcy of Lehman Brothers' stock price fluctuations, banking institutions have faced considerable consequences in the banking sector and financial markets. Falls in prices threaten market participants, investors and policy makers. In the post-crisis period of the recent global financial crisis, stock prices of the banking sector can be extremely volatile. In extreme volatile periods of stock prices, market participants might suffer large losses, revealing the importance of the trading volume effect on banks' stock return volatility.

This work investigated the direct impact of the trading volume of Greek banking institutions on stock return volatility considering the four major banking institutions in Greece: (i) Alpha bank, (ii) Eurobank, (iii) National Bank of Greece and (iv) Piraeus Bank. The importance of the trading volume has been highlighted by the literature which acted as a moderator for the conditional price volatility in financial markets; in other words, liquidity increases via trading volume may improve the climate in harsh times. We focused on daily measures of volatility estimated from intraday data to explain volatility changes over time for a large period of twenty years (5 January 2001 to 30 December 2020). We measured the stock return volatility using four different volatility estimators to check the consistency of the trading volume impact on stock return volatility. The results reveal a direct impact of trading volume on stock return volatility mainly in all quantiles (this is in line with the theory). These findings are recommended to market participants, policy makers and practitioners dealing with the Greek financial market. Further research should be focused on the performances of different estimators of volatility in the banking sector, and this approach may provide a more complete idea of the volatility that outperforms in the banking sector.

**Author Contributions:** The four authors contributed equally to this paper. All authors have read and agreed to the published version of the manuscript.

**Funding:** This research was funded by Greece and the European Union (European Social Fund-ESF) through the Operational Programme "Human Resources Development, Education and Lifelong Learning 2014–2020" and "The APC was funded by the authors".

**Acknowledgments:** This research is co-financed by Greece and the European Union (European Social Fund- ESF) through the Operational Programme "Human Resources Development, Education

and Lifelong Learning 2014–2020" in the context of the project "Minimize transmission of systemic banking risk through automatic short-term intervention mechanisms (intra-day circuit breaker). A study on the Greek banking system and capital controls" (MIS 5047178).

**Conflicts of Interest:** The authors declare no conflict of interest.

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
