# Peer review of "Does Trading Volume Drive Systemic Banks’ Stock Return Volatility? Lessons from the Greek Banking System"

_ijfs, doi:10.3390/ijfs9020024_

Round 1
Reviewer 1 Report
The paper does not respond with sufficient scientific rigor for this type of journal. It would need a major review and structuring to achieve the structure that is requested in this class of publications and scientific magazines.
Some of the aspects to improve are listed below.
In the section on statistical data and results, several issues are appreciated. First, the database from which the data is obtained is not sufficiently reliable. Databases such as Thomson Reuters would be more suitable for drawing conclusions in this type of study and journal.
The explanation shown in the text of the different tables is reiterative. It would not be necessary to re-express the results that are already in the tables. In any case, reflect an interpretation of them.
Table 2 should be in the empirical results section and not in descriptive data and results. It should be next to table 3, 4 and 5 which are of the same category.
In the methodology section, in reference to the format, the size of the parameters must be unified when they are defined within the text, for example (lines 197-199), although it occurs on many occasions which must be reviewed.
The conclusions section should be the most important of the research and it is not given the importance it deserves. It consists only of a paragraph which half is to reiterate what is indicated in the abstract. Rewrite and improve the conclusions.
In summary, the paper must be carefully reviewed and properly structured to be at the level of this journal.
Reviewer 2 Report
Dear authors,
thank you for your effort. I have several recommendations to enrich your study.
- Fully depersonalize the manuscript (do not use we, our, at all).
- Abstract: set the kind of the research used and determine main aim of the manuscript.
- Introduction: Samitas and Polyzos (2016) and Stournaras (2019), these references have larger font than others, check it, please. I suggest to include in your literature review also these references to extend the view:
Musa, H., Natorin, V., Musova, Z., & Durana, P. (2020). Comparison of the efficiency measurement of the conventional and Islamic banks. Oeconomia Copernicana, 11(1), 29-58.
Balcerzak, A. P., Kliestik, T., Streimikiene, D., & Smrcka, L. (2017). Non-parametric approach to measuring the efficiency of banking sectors in European Union Countries. Acta Polytechnica Hungarica, 14(7), 51-70. - Data and descriptive statistics: explain why did you used Jarque-Bera test to prove normality.
- Methods: use the same style for all formulas, explain all variable symbols used.
- Discussion: it is crucial chapter of the paper, add it and compare your empirical results to the similar studies,
- Conclusions: add limitations and further ways of the research.
Good luck in your future work.
I will recommend publishing your manuscript in scientific IJFS after your corrections.
Round 2
Reviewer 1 Report
After making the proposed corrections, the paper has a higher scientific quality. Congratulations.Reviewer 2 Report
Dear authors,
thank you for your precise work, I am fully satisfied with your corrections.
I recommend publishing.
Good luck in your future research.